# Implications of Senescent Cell Burden and NRF2 Pathway in Uremic Calcification: A Translational Study

**DOI:** 10.3390/cells12040643

**Published:** 2023-02-17

**Authors:** Jonas Laget, Sam Hobson, Karen Muyor, Flore Duranton, Irene Cortijo, Piotr Bartochowski, Bernard Jover, Anne-Dominique Lajoix, Magnus Söderberg, Thomas Ebert, Peter Stenvinkel, Àngel Argilés, Karolina Kublickiene, Nathalie Gayrard

**Affiliations:** 1RD-Néphrologie, 34090 Montpellier, France; 2Biocommunication in Cardio-Metabolism (BC2M), University of Montpellier, 34090 Montpellier, France; 3Division of Renal Medicine, Department of Clinical Science, Intervention and Technology, Karolinska Institutet, 141 52 Stockholm, Sweden; 4Pathology, Clinical Pharmacology and Safety Sciences, R&D AstraZeneca, 431 50 Gothenburg, Sweden; 5Medical Department III–Endocrinology, Nephrology, Rheumatology, University of Leipzig Medical Center, 04109 Leipzig, Germany

**Keywords:** vascular calcification, kidney failure, NRF2, senescence, subtotal nephrectomy

## Abstract

Increased senescent cell burden and dysregulation of the nuclear factor erythroid 2–related factor 2 (NRF2) pathway have been associated with numerous age-related pathologies; however, their role in promoting vascular calcification (VC) in chronic kidney disease (CKD) has yet to be determined. We investigated whether senescence and NRF2 pathways may serve as drivers of uremia-induced VC using three complementary approaches: a novel model of induced VC in 5/6-nephrectomized rats supplemented with high phosphate and vitamin D; epigastric arteries from CKD patients with established medial calcification; and vascular smooth muscle cells (VSMCs) incubated with uremic serum. Expression of p16^Ink4a^ and p21^Cip1^, as well as γ-H2A-positive cells, confirmed increased senescent cell burden at the site of calcium deposits in aortic sections in rats, and was similarly observed in calcified epigastric arteries from CKD patients through increased p16^Ink4a^ expression. However, uremic serum-induced VSMC calcification was not accompanied by senescence. Expression of NRF2 and downstream genes, *Nqo1* and *Sod1*, was associated with calcification in uremic rats, while no difference was observed between calcified and non-calcified EAs. Conversely, in vitro uremic serum-driven VC was associated with depleted NRF2 expression. Together, our data strengthen the importance of senescence and NRF2 pathways as potential therapeutic options to combat VC in CKD.

## 1. Introduction

Chronic kidney disease (CKD) is a highly debilitating non-communicable disease [1], affecting close to 800 million patients worldwide in 2017 [2]. With decreasing estimated glomerular filtration rate (eGFR), accelerated cardiovascular aging translates to an increased risk of cardiovascular disease (CVD) and mortality [3,4]. This is due to, in part, increased vascular calcification (VC) [5,6] deposited in the intima and media layers of blood vessels [7]. The pathophysiology of VC in CKD has yet to be fully understood and therapies targeting the accelerated calcification process are warranted despite recent identification of inhibitors such as vitamin K, inositol phosphate analogues or “calcification blocking factor” [8,9,10,11]. Evidence suggests that senescence and nuclear factor erythroid 2–related factor 2 (NRF2), two different yet connected pathways, could be involved in VC in CKD [12]; thus, a possible therapeutic strategy is to target such common pathways underpinning the aging process—an approach that has been successful in pre-clinical animal models of disease [13].

Senescence is the state of irreversible cell cycle arrest [14]. Although beneficial in development and wound healing [15,16], increased senescent cell burden is associated with age-related diseases [14] and constitutes one of nine proposed hallmarks of aging [17]. Senolytics, compounds that specifically target senescent cells, can ameliorate disease-specific dysfunctions [18]. Current practices to detect senescence rely on the combination of multiple markers such as the expression of cyclin-dependent kinase inhibitors p16^Ink4a^ and p21^Cip1^ (contribute to G1/S cell cycle blockade), serine 139 phosphorylated H2A histone family member X (γ-H2AX), presence of Senescence Associated Secretory Phenotype (SASP) and lysosomal senescence-associated beta-galactosidase (SA-β-gal) activity [14,19,20].

The NRF2 pathway has been termed the ‘master regulator’ of antioxidant genes [21]. NRF2 is a transcription factor ubiquitinated by the Kelch-like ECH-associated protein 1 (Keap1)–cullin3 (Cul3) complex under basal conditions [22]. However, in response to stress, NRF2 ubiquitination is halted following a conformational change in Keap1, allowing NRF2 to translocate to the nucleus where it binds to conserved antioxidant response element (ARE) sequences, thereby regulating transcriptional activity of a plethora of genes with antioxidant functions [23]. This suggests NRF2 is a potential therapeutic target in conditions associated with impairment in oxidative equilibrium, such as aging [24].

Despite representing two distinct pathways, a bidirectional relationship between NRF2 and senescence has been reported [25,26,27]. To further elucidate senescence and NRF2 functions in the context of uremia-induced VC, we investigated their expression in aortic sections of 5/6-nephrectomized (SNX) rats supplemented with phosphate and vitamin D, a novel model of CKD and VC. We also investigated expression patterns in human epigastric arteries and an in vitro uremic serum-induced calcification model to gain a comprehensive understanding of their regulation in established (human) and induced (rats and cultured cells) calcification.

## 2. Materials and Methods

See Appendix A and Methods for more details.

### 2.1. Animal Model of Uremic VC: Tissue Sampling and Blood Parameters

Twenty-four male Sprague Dawley (OFA) rats (Charles River Laboratories, Les Oncins, France) were housed in specific facilities (permit D3417225). Eighteen eight-week-old rats underwent SNX in a one-step procedure, and six rats underwent sham operation (control group) [28]. Briefly, under isoflurane (2% in O_2_) volatile anesthesia and buprenorphine analgesia (Bupaq, 0.1 mg/kg, subcutaneous injection), a ventral laparotomy was used to expose the kidneys. The right kidney was excised and two of the three branches of the left renal artery were ligated. Six sham-operated rats followed laparotomy and manipulation of the two kidneys without renal mass reduction. Rats were left for eight weeks without manipulation for the establishment of CKD pathophysiology (Appendix A). Twelve SNX rats (≈500 g weight) then initiated a four-week VC diet—standard diet supplemented with high phosphate (1.2% P, 1.04% Ca, SAFE) and 1-hydroxy-vitamin D in food pellet (Alfacalcidol, LEO pharma, Voisins-le-Bretonneux, France) at a dose of 0.1 µg/day (SNX 0.1 group, n = 6) or 0.4 µg/day (SNX 0.4 group, n = 6)—while sham-operated rats (control, n = 6) and part of the SNX group (SNX, n = 6) were maintained on a standard diet (A04 0.55% P, 0.73% Ca, SAFE, Augy, France). After four weeks, urine production was obtained and arterial pressure was measured (carotid catheterization under 2% isoflurane anesthesia) before blood and organs were harvested after euthanasia (200 mg/kg pentobarbital, intravenous injection). Creatinine, urea, total protein, phosphate, and calcium concentrations were measured in plasma and urine on a COBAS automated analyzer (Roche Diagnostics, Meylan, France). No animals were excluded during the analysis. Animals were housed in groups of 4 (one animal of each group in the same cage). Data collection was blinded to reduce bias.

### 2.2. Observational Clinical Study and In Vitro Assay: Patients and Sampling

For all patients, inferior epigastric arteries and blood were obtained at the time of living-donor kidney transplantation (LD-KTx) and stored for analysis. In short, epigastric arteries from 16 kidney failure patients were scored for calcification by a trained pathologist and graded from 0 (no calcification) to 3 (severely calcified). CKD was classified as abnormalities of kidney function that persist for >3 months, i.e., low glomerular filtration rate (GFR) or elevated urine albumin should be detectable for at least 90 days [29], while kidney failure referred to patients with a GFR < 15 mL/min/1.73 m^2^ that require renal replacement therapy [30]. Patients with severely calcified (n = 8) and non-calcified (n = 8) vessels were matched for age, sex, CVD, statin usage and presence of diabetes (Table 1). Extremes, i.e., grades 0 and 3, were chosen based on sample availability after matching, while male subjects were included to complement animal experiments and mitigate potential sex-related differences [31,32]. Data were also obtained for % calcification in epigastric artery vessel segments (for only 10/16 patients), measured semi-quantitatively in a previous study [33,34]. hsCRP and creatinine quantification were performed by routine laboratory tests. The 8-OHdG was determined using High Sensitivity 8-OHdG ELISA Assay Kit (JaICA). Patient epigastric artery segments were used for subsequent protein and RNA expression analysis and uremic serum from patients with calcified vessels was used for in vitro calcification experiments. All experiments were performed in accordance with the Declaration of Helsinki.

### 2.3. Histology

A total of 4–5 µm sections of kidney, heart, or arteries was used for determination of fibrosis (0.1% picrosirius red). Von Kossa (silver nitrate plus nuclear fast red) was used to study VC in human epigastric arteries, and thoracic aorta (TA) and aortic arch (AA) in rats. Calcium content of thoracic aortas was determined by the o-cresolphthalein complexone method (Calcium Colorimetric Assay Kit, Abcam, Paris, France). To obtain an integrated estimate of VC in rats, a score accounting for two tissues (aortic arch and thoracic aorta) and two methods (von Kossa and Ca content) was established (Appendix A) according to the following formula:$$\mathrm{VC}\;\mathrm{score}=\frac{1}{2}\left(\frac{\mathrm{stained}\_\mathrm{AA}}{\mathrm{Mean}\;(\mathrm{stained}\_\mathrm{AA}\;(\mathrm{SNX}\;0.1/0.4))}+\frac{1}{2}\left(\frac{\mathrm{stained}\_\mathrm{TA}}{\mathrm{Mean}\;(\mathrm{stained}\_\mathrm{TA}\;(\mathrm{SNX}\;0.1/0.4))}+\frac{\mathrm{Ca}\_\mathrm{TA}}{\mathrm{Mean}\;(\mathrm{Ca}\_\mathrm{TA}\;(\mathrm{SNX}\;0.1/0.4))}\right)\right)$$

VC score, only calculated for SNX 0.1 and SNX 0.4, was used to test for correlation with other biological parameters in rats.

Immunohistochemistry (IHC) (Vectastain ABC kit, Vector Labs, Newark, CA, United States) and immunofluorescence (IF) were performed on TAs and EAs with anti-p16^Ink4a^, anti-p21^Cip1^, anti-NRF2, anti-γH2AX, anti-osteopontin (OPN), anti-RUNX family transcription factor 2 (Runx2) and anti-alpha smooth muscle actin (α-SMA) primary antibodies and appropriate secondary antibodies. Fluorescence was observed with Axio-imager Z Apotome (Zeiss, Jena, Germany) using Montpellier MRI imaging platform or Axio-observer Z1 microscope (Zeiss, Jena, Germany).

### 2.4. mRNA Expression

Total RNA extraction was performed for TA, human epigastric arteries and VSMCs. RNA quality was examined and qPCR performed after first strand DNA synthesis. For TA, data were analyzed with LightCycler 96 software 1.1 and mRNA expression was quantified as a Relative Ratio using three housekeeping genes: β-act, Gusb, and Hprt1. For epigastric arteries and VSMCs, data were analyzed using QuantStudio 7 Flex Real-Time PCR System (Applied Biosystems, Waltham, MA, USA) and relative expression was calculated with the 2ΔΔCt method, using GAPDH as reference. For human samples, mRNA expression in the form of Ct (housekeeping gene)/Ct (gene of interest) was also calculated to correlate with other genes of interest, % calcification of vessels and hsCRP. See primers sequences in Appendix A.

### 2.5. In Vitro Assay

Pooled uremic serum was obtained from the 8 patients presenting severe epigastric artery calcification (Table 1). Non-CKD control serum (n = 8) from the PRIMA-control cohort consisted of an age- and sex-matched population randomly selected from the Stockholm Region, Sweden, by the Statistics Bureau of Sweden—a government agency [35]. Human aortic VSMCs (ATCC) from a healthy male donor were incubated between passages 4–77 with one of the following conditions: control (DMEM), high phosphate (2.5 mmol/L sodium phosphate), non-CKD control serum (2.5 mmol/L sodium phosphate + 10% pooled non-CKD control serum) or uremic serum (2.5 mmol/L sodium phosphate + 10% pooled uremic serum). Cells were incubated for 7 days (37 °C, 5% CO_2_). On day 7, cells were harvested for calcium content assay using IRDye^®^ 800 CW BoneTag™, (Licor, NE, USA) and qPCR analysis, or fixed for SA-β-Gal staining (Sigma-Aldrich, CA, USA). For the calcium content assay, fluorescent signals were detected using the Odyssey CLx Infrared imaging system (Licor, NE, USA). Readouts were then normalised for protein content using BCA protein assay (Abcam, Amsterdam, The Netherlands). 

### 2.6. Statistics

Statistical analysis was performed using GraphPad Prism v8.0.2 (San Diego, CA, USA). For categorical variables, Chi-squared test was used to compare groups. For all continuous variables, Kolmogorov–Smirnov test was used to test for normality. Group comparisons were performed using independent t-test for normally distributed data or Mann–Whitney test for non-parametric data. Comparisons between three or more groups were performed using one-way ANOVA followed by Tukey’s post hoc tests. Univariate correlations were performed using Pearson’s rank-order method or Spearman’s rank correlation coefficient depending on data distribution. *p*-values < 0.05 were considered statistically significant.

### 2.7. Study Approval

All animal experiments were performed according to the European Parliament Directive 2010/63/EU (N° CEEA-00322.03) and approved by the local ethics committee for animal experimentation of Languedoc-Roussillon (CEEA-LR, n°036, #18348). This study is reported in accordance with ARRIVE guidelines where applicable [36]. Adult patients undergoing LD-KTx at Karolinska University Hospital were invited to participate after written informed consent (approved by Regional Ethical Review Board in Stockholm) [35].

## 3. Results

### 3.1. Validation of the Pre-Clinical Uremic VC Model

SNX groups presented typical features of CKD such as proteinuria and decreased creatinine clearance (Table 2). Plasma calcium and phosphate [Ca × P] product was increased in SNX 0.4 group (*p* < 0.05). An increase in pulse pressure (*p* < 0.0001) was observed in SNX 0.4 animals (Table 2) and the left ventricle weight (*p* < 0.0001) was higher in the three SNX groups with presence of heart fibrosis (*p* < 0.05, Table 2). The final weight did not differ between control, SNX, SNX 0.1, and SNX 0.4 groups at the end of the experiment (Table 2).

VC measured by von Kossa staining and Ca content was clearly increased in aortic arches and thoracic aortas in SNX 0.4 group (Figure 1a–c, *p* < 0.05 vs. SNX and controls), and the same trend was observed in SNX 0.1 group (ns vs. SNX and controls). VC occurred in 50% of SNX 0.1 and 83% of SNX 0.4 animals (in three and five rats, respectively), while it was absent in SNX and control groups (Figure 1). We found the vitamin D dose increased the incidence of VC rather than its intensity. Next, we developed a VC score based on VC estimates from different tissues (thoracic aorta and aortic arch) and techniques (von Kossa staining and calcium content) to use as an overall VC intensity scale (see Georgiadis et al., study for a comparable approach in kidney failure patients) [37]. The VC score was significantly increased in SNX 0.4 group (Figure 1d, *p* < 0.01) and strongly correlated with pulse pressure (Figure 1e, n = 12, *p* < 0.01).

### 3.2. VSMC Osteoblastic Transition and Thoracic Calcification in Uremic Rats

mRNA expression of pro-calcifying RUNX family transcription factor 2 (*Runx2*, *p* < 0.01), Bone Morphogenetic Protein 2 (*Bmp2*, *p* < 0.05) and Solute Carrier Family 20 Member 1 (*Pit-1*, *p* < 0.01) positively correlated with VC in TA, while protective Solute Carrier Family 20 Member 2 (*Pit-2*, *p* < 0.01) negatively correlated with calcification (Table 3, Appendix A). No difference was observed between the four groups at the mRNA level, mainly due to a high dispersion of data in SNX 0.1 and SNX 0.4 rats, of which thoracic aortas with and without VC were included (Figure 1). Using IF, we observed specific RUNX2 protein expression close to calcium deposits (Appendix A). Conversely, alkaline phosphatase (*Alpl*, *p* < 0.01) and matrix metalloproteinase (*Mmp2*, *p* < 0.01) were negatively correlated to VC (Figure 1f, Table 3). In the same way, mRNA expression of VC inhibitors, Matrix Gla protein (*Mgp*, *p* < 0.01), *Opn* (*p* < 0.001), and Ectonucleotide pyrophosphatase/phosphodiesterase 1 (*Enpp1*, *p* < 0.05) positively correlated with VC (Figure 1g, Table 3) and significantly increased in SNX 0.4 versus control groups for *Mgp* (*p* < 0.01) and *Opn* (*p* < 0.05). OPN expression was closely associated with calcification (Appendix A). These data suggest counteracting mechanisms trying to tackle VC. Alpha smooth muscle actin (*α-SMA*), a classic marker of contractile vascular smooth muscle cells (VSMCs), was decreased in the SNX 0.1 (*p* < 0.05) and SNX 0.4 (*p* < 0.05) groups and negatively correlated with VC score (*p* < 0.001, Figure 1g, Table 3).

### 3.3. Increased Senescent Cell Burden in Calcified Rats

We assessed the expression of senescence markers p16^Ink4a^, p21^Cip1^ and γH2AX in rat TAs. p16^Ink4a^ expression at mRNA and protein level (Figure 2a,c,d) positively correlated with VC (*p <* 0.05 for mRNA and *p* < 0.01 for protein expression, Table 3 and Figure 2e, respectively). p16^Ink4a^ protein expression was observed in vascular cells’ nuclei in the calcified area (Figure 2c and Figure 3d). Protein expression of p21^Cip1^ was increased in SNX 0.4 versus controls (*p* < 0.05) and positively correlated with VC score (*p* < 0.05, Figure 2c–e). At the mRNA level, p21^Cip1^ was negatively correlated with VC (Table 3, *p* < 0.05), suggesting important post-transcriptional regulation of p21^Cip1^ expression [38]. p21^Cip1^ protein was expressed close to calcium deposits (Appendix A, Figure 2c). In addition, mRNA expression of *p53* was increased in SNX 0.1 (*p* < 0.05) and SNX 0.4 (*p* < 0.001) groups (Figure 2a). We also investigated the production of distinct cytokines and chemokines contributing to the SASP, and found that Interleukin 1 beta (*Il-1β)* (*p* < 0.05) and Monocyte Chemoattractant Protein-1 (*Mcp-1*, *p* < 0.01) mRNA expression was increased in the SNX 0.4 group, and *Mcp-1* expression correlated with VC score (*p* < 0.05, Figure 2b, Table 3).

Senescent cells display DNA damage response components such as γH2AX. Using IF, we found an increase in γH2AX-positive cells close to calcium deposits in SNX 0.4 group versus control and SNX (*p* < 0.01 for both, Figure 3a,b), which strongly correlated with the level of VC (*p* < 0.001, Figure 3c). We observed that γH2AX-positive cells also expressed p16^Ink4a^ (Figure 3d), thus displaying different markers of senescent cells. In addition, some γH2AX-positive cells also expressed α-SMA, indicating contractile VSMC origin of senescent cells in calcified TAs (Appendix A).

### 3.4. Oxidative Stress Regulation through NRF2 Pathway in Calcified Rats

NRF2 expression was increased in the SNX 0.4 group versus controls at both the mRNA (Figure 4a, *p* < 0.05) and protein (Figure 4b,c, *p* < 0.01) levels, and positively correlated with VC (*p* < 0.05 for mRNA and *p* < 0.001 for protein, Table 3, Figure 4d). mRNA expression of NAD(*p*)H quinone dehydrogenase 1 (*Nqo1*), a downstream target of NRF2, increased with VC score (*p* < 0.01, Table 3). mRNA expression of Glutathione peroxidase 4 (*GPx4*), also regulated by NRF2, was increased in the SNX 0.4 group (Figure 4a, *p* < 0.01). IF revealed that NRF2 was expressed in vascular cells close to calcium deposits and localized in the nucleus (Figure 4e), suggesting NRF2 translocation from the cytoplasm. 

### 3.5. Increased Senescent Cell Burden, but Not NRF2 in Human Calcified Arteries

We designed a follow-on study to compare senescence and NRF2 expression patterns in eight kidney failure patients with severely calcified vessels vs. a matched group (Table 1) of eight non-calcified kidney failure patients, isolated at LD-KTx. As expected, the mean coronary artery calcification (CAC) score (463 vs. 0 AU; *p* < 0.05; Table 1) was significantly higher in calcified kidney failure patients.

In calcified epigastric arteries (Appendix A), *RUNX2* mRNA expression was significantly increased compared to non-calcified vessels (*p* < 0.01; Appendix A), and expression tended to correlate with the level of calcification (rho = 0.63, *p* = 0.06Appendix A). *p16*^Ink4a^ mRNA expression was significantly increased in the calcified group (*p* < 0.001, Figure 5a) and correlated with the degree of calcification (rho: 0.64, *p* < 0.05; Figure 5a). Similar to rat calcified thoracic aortas, calcification was present in the media of epigastric arteries and tissue disruption was observed in calcified areas after von Kossa staining (Figure 5a). p16^Ink4a^ protein expression (*p* < 0.05, Figure 5c,d) was also elevated in the media layer of the calcified group. Furthermore, *p16*^Ink4a^ expression correlated with *ALPL* (rho: 0.54, *p* < 0.05; Figure 5a), *RUNX2* (rho: 0.85, *p* < 0.001; Figure 5a) and high-sensitivity C-reactive protein (hsCRP) (rho: 0.64, *p* < 0.01; Figure 5a). The univariate correlation between p16 and p21 was not significant (*p* > 0.05; data not shown). No difference was observed between calcification groups for *p21*^Cip1^ mRNA expression (*p* > 0.05, Figure 5b). Still, p21^Cip1^-positive cells were present in calcified vessels, although not significantly increased (*p* > 0.05, Figure 5e–f). 

8-Hydroxydeoxyguanosine (8-OHdG), a marker of oxidative stress, was significantly increased in the severely calcified patients (*p* < 0.05; Figure 6a). However, NRF2 (mRNA and protein) and *NQO1* (mRNA) were unchanged between the calcified and non-calcified groups (both *p* > 0.05, Figure 6a,d), while *SOD1* mRNA expression was significantly decreased in the calcified group. *NRF2* gene expression did not correlate with other parameters evaluated, including hsCRP, osteogenic genes, % calcification, *p16*^Ink4a^ or *p21*^Cip1^ expression (Figure 6b,c).

### 3.6. NRF2 Dysregulation In Vitro

VSMC calcification was increased when osteogenic media were supplemented with uremic serum, compared to DMEM, high-phosphate and non-CKD control serum groups (all *p* < 0.0001; Figure 7a). Gene expression analysis was then performed for the uremic serum condition since a pro-calcifying effect was observed. We found *ALPL* gene expression increased in the presence of uremic serum (*p* < 0.0001 vs. controls; Figure 7b), while *RUNX2* expression remained unchanged. Msh Homeobox 2 (*MSX2*) gene expression was elevated by uremic serum, however, it did not reach statistical significance (Figure 7b). Uremic serum-induced calcification was associated with lower *NRF2*, *SOD1* and *NQO1* gene expression (all *p* < 0.01 vs. osteogenic control; Figure 7c), however, it was not associated with changes in senescence markers *p16*^Ink4a^ and *p21*^Cip1^ (both *p* > 0.05; Appendix A) or SA-β-Gal staining (*p* > 0.05; Figure 7d).

## 4. Discussion

This study demonstrates that VC is associated with senescence and modified NRF2 signaling, adding new insights of translational mechanisms related to uremia-induced early vascular aging. In an induced in vivo model of VC, we showed the presence of senescent cells in the TA, and its abundance correlated with the degree of calcification, in addition to the presence of SASP components. This is in line with increased p16^Ink4a^ expression in epigastric arteries from kidney failure patients presenting established calcification. The present study also illustrated that “protective” NRF2 signaling is increased in the presence of induced calcification in rats [39,40]. In contrast, despite preservation of NRF2 signaling in calcified epigastric arteries from kidney failure patients, its target *SOD1* was decreased, potentially participating in the established vessel calcification. We also observed that depleted NRF2 is associated with uremic serum-induced calcification in vitro, further highlighting differences in NRF2 regulation between different models of uremia-induced VC. 

SNX rodents maintained on a high-phosphate diet rarely present VC, even 20 weeks after SNX [41,42]. The idea to provoke ectopic calcification in SNX animals using high phosphate and/or vitamin D is not new and such models have been previously validated.^41^ Based on the literature, we developed a surgery and diet-induced model that presented VC in the media layer, while limiting stressful interventions such as oral gavage or intraperitoneal injections and allowed sufficient time for CKD to progress before inducing calcification. Animals did not present overt signs of diet-induced toxicity and the two doses of alfacalcidol (i.e., 1-hydrocholecalciferol, up to 0.8 µg/kg) resulted in similar levels of VC, but the incidence increased with the higher dose. 

Typical genes involved in VSMCs osteoblastic transition and calcification, e.g., *Runx2*, *Bmp2*, *Pit-1* [43], etc., were elevated in calcified TA, while *α-SMA* and *Pit-2* expression decreased [44,45]. We also found mRNA expression of multiple inhibitors of VC in the thoracic aorta, such as *Mgp*, *Opn*, and *Enpp1*, positively correlated with calcification, and *Alpl*, which promotes VC, decreased [46,47], suggesting compensatory gene expression mechanisms developed to counteract induced VC. Increased expression of *Nrf2* and its downstream targets in calcified TAs indicates another protective mechanism in our model and validates the findings of Jin et al. who observed upregulation of NRF2 in another model of uremic VC [48]. The upregulation of NRF2 signaling was confirmed at the protein level; expression was strongly correlated with VC intensity, and NRF2 translocation to the nucleus was observed close to calcified areas. As with Opn and Mgp, NRF2 overexpression in our model of artificially induced calcification was not causal but in response to VC. This observation has been previously shown in the context of Parkinson’s Disease where NRF2 and downstream target SQSTM1 were overexpressed in amyloid precursor protein- and tau-injured neurons [49,50]. In contrast to rodent data, mRNA and protein expression were preserved in calcified and non-calcified human epigastric arteries, with decreased *SOD1*, albeit with a marked increase in oxidative stress. Time exposed to the uremic milieu may account for this observation, in addition to more complex regulatory systems in humans under uremic conditions. In contrast to animal models, kidney failure patients have been exposed to a toxic uremic environment for years [51]. The literature shows that uremic toxins, inflammation and oxidative stress downregulate NRF2 [52,53]. Thus, exhaustion of a counter-regulatory increased expression of NRF2 and its related antioxidant targets from prolonged oxidative milieu may facilitate early vascular aging and VC. We also cannot rule out that preserved expression is indicative of NRF2 not contributing to calcium—phosphate deposition in the human setting, despite several studies suggesting aging is associated with depleted NRF2 expression [54]. In vitro, VSMC calcification in the uremic context was associated with reduced mRNA expression of *NRF2*, and downstream targets *NQO1* and *SOD1.* This may suggest the presence of circulating factors in the serum of calcified uremic patients exhausting NRF2 signaling, such as NRF2 repressor Bach1 [55]. Hyperphosphatemia-induced calcification is associated with NRF2 depletion in vitro, and calcium—phosphate deposition can be alleviated with NRF2 agonists, such as tert-butylhydroquinone and dimethyl fumarate [39,40]. Discordant findings between different models in relation to NRF2 expression might be explained by our inability to recapitulate the aging phenotype in SNX rats, as well as the use of high-vitamin D and high-phosphate supplementation. To the best of our knowledge, a study investigating the effects of uremic serum-induced calcification in the context of NRF2 activity has not been previously reported. Our data support the complexity of NRF2 regulation in disease and indicate that timing of therapeutic intervention needs careful consideration [56]. This is perhaps best exemplified in the setting of cancer, where overexpression is associated with tumorigenesis and resistance to chemotherapy [57]. Diabetic nephropathy animal models also show conflicting results with both NRF2 downregulation and upregulation reported. It has been postulated that CKD severity may partially explain differences in NRF2 regulation [58]. However, the absence of clinical biomarkers that determine duration of action and target engagement has slowed clinical development, and currently relies on the expression of NRF2 and downstream targets [59]. Taken together, our data show that ‘protective’ NRF2, increased in an in vivo model of induced VC, was preserved in ESRD patients with VC and uremic serum from these patients may contain interacting factors explaining depleted NRF2 in vitro, confirming the reputed link between NRF2 and VC. Future studies are needed to further describe the effect of NRF2 activation and repression in VC and investigate a potential causal relationship.

We also interrogated senescence in relation to VC, which so far has been predominantly studied in the in vitro setting [60,61]. Senescence-associated VC was also investigated in adenine rats [62,63]; however, studies in the SNX model, which is closer to CKD pathophysiology [64], are lacking. The presence of senescent cells in calcified TAs was confirmed with detection of p16^Ink4a^, p21^Cip1^ and γ-H2AX, all of which positively correlated with the level of calcification in rats. In addition, mRNA expression of pro-inflammatory SASP cytokines increased in calcified vessels. SASP components contribute to senescence-related inflammation, and the increased expression of inflammatory cytokines could favor VSMC differentiation into osteoblastic-like cells and vascular calcification [65]. Calcium phosphate crystals can elicit a pro-inflammatory response from macrophages leading to a positive feed-back loop between calcification and inflammation [66,67]. Cells positive for both p16^Ink4a^ and γ-H2AX were closely associated with calcification, illustrating the link between DNA damage response, senescence and VC [61]. In kidney failure patients, p16^Ink4a^ mRNA and protein expression were increased in calcified versus non-calcified epigastric arteries, however, no differences were observed for p21^Cip1^. The presence of senescent cells at the site of insult in atherosclerotic plaques and calcified arteries may support our findings [33,68]. p21^Cip1^ expression has been associated with transient cell cycle arrest and may not be continually expressed in senescent cells [69]. It is important to stress that p16^Ink4a^ and p21^Cip1^ may also have differential roles in the senescent-cell-cycle arrest process. For example, p21^Cip1^ may be responsible for the initial inactivation of cyclin-associated kinase activity, while p16^Ink4a^ is important for maintenance of senescent-cell-cycle arrest [70]. Since p16^Ink4a^/p21^Cip1^ expression differences were not consistent between groups in SNX rats and kidney failure patients, we speculate that the potential distinctive roles of these two proteins may explain these findings. Our results also corroborate with Sanchis et al.’s study that shows VSMCs in vessels from children on dialysis presented higher levels of senescence markers (p16^Ink4a^, p21^Cip1^), persistent DNA damage (γ-H2AX) and were more prone to osteogenic differentiation and calcification in vitro [71]. We did not observe differences in senescent cell abundance in vitro after application of uremic serum for induction of calcification. Although we opted for a seven-day incubation period based on similar experimental protocols to induce VSMC calcification [72,73,74], it may be argued that a longer incubation time would be needed since senescence may take between 10 days and 6 weeks to be induced in vitro [75].

Therapeutics targeting the calcification process in CKD are limited. While phosphate binders and calcimimetics are viable treatment options for patients with abnormal phosphate metabolism, unequivocal evidence of these compounds reducing major cardiovascular events in randomized controlled trials is lacking [8,76]. Therapeutically targeting central aging pathways may be more effective in resolving or delaying the onset of the multiple burden of lifestyle diseases simultaneously, rather than approaching each disease as a single entity. While clinical trials investigating the effects of NRF2 agonists (NCT04608903) and senolytics (NCT02848131) in CKD patients are ongoing, to the best of our knowledge, their potential effects on VC or early vascular aging are not being investigated.

Strengths of our study include the translational design using a novel rat model of uremic VC, arteries from kidney failure patients and in vitro experiments. We also implemented a plethora of different experimental tools to assess senescence, since a universal marker has yet to be identified. We also acknowledge limitations of our study, such as the necessity to better assess the time-dependent regulation of calcification, senescence and NRF2 signaling pathways, which could not be envisaged in the present work. In addition, our study is mainly descriptive; future work will address causation between NRF2 or senescent cell burden with VC and combine treatments targeting both signaling pathways for potential additive/synergistic effects. However, as a first step, we aimed to characterize the presence of senescent cells and NRF2 expression in our novel animal model.

In summary, for the first time we have shown in a novel animal model of uremic calcification that increased senescent cell burden and dysregulated NRF2 expression are linked to medial VC, often at the site of insult. These findings are further supported in calcified epigastric arteries from kidney failure patients in the case of senescence. The observation of preserved NRF2 expression in calcified vs. non-calcified uremic arteries further strengthens the importance of the sequence of events during early vascular aging, and stresses the need to understand differences between induced and long-time established calcification. Our study gives a greater translational insight into how two central pathways contribute to VC and provides a basis for interventional studies. 

## Figures and Tables

**Figure 1 cells-12-00643-f001:**
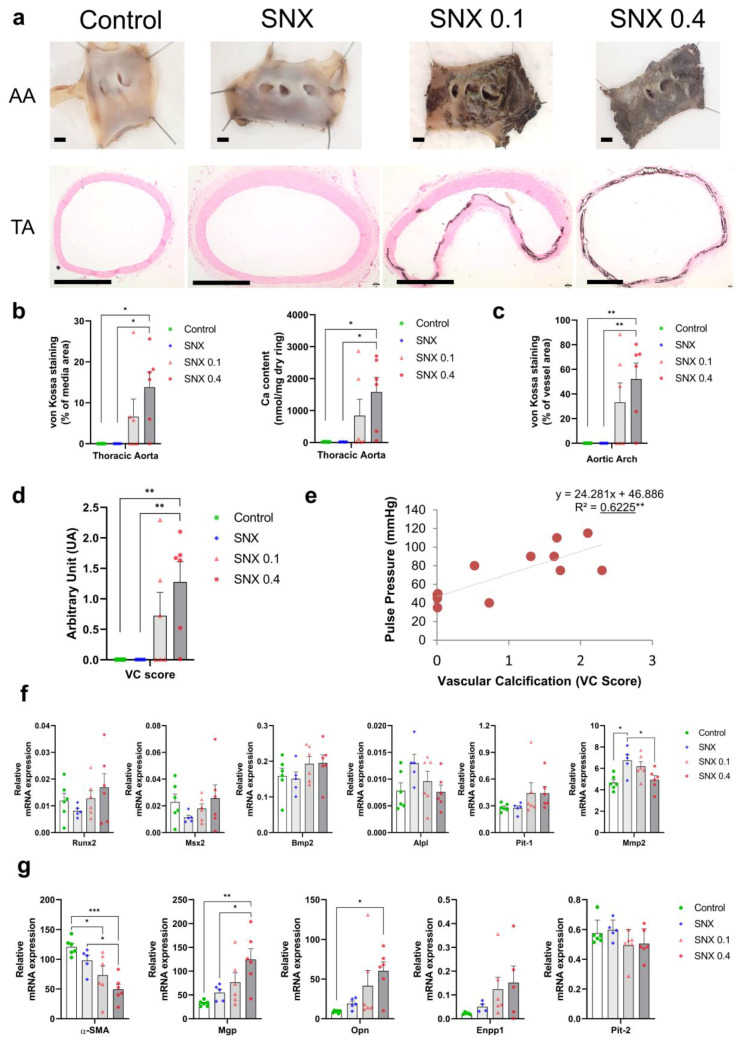
Analysis of calcification in the aortic arch and thoracic aorta, and changes in mRNA expression in thoracic aorta, of uremic Sprague Dawley rats after diet supplementation with high phosphate (1.2%) and 0.1 µg/day (SNX 0.1) or 0.4 µg/day (SNX 0.4) vitamin D. (**a**) Representative images of von Kossa staining in aortic arch (AA) and thoracic aorta (TA). For AA samples, after von Kossa protocol, the vessels were cut longitudinally and fixed with needles to visualize staining on the inner side. Scale bar = 1 mm. (**b**) Calcification was quantified as the percentage of media area stained by von Kossa and calcium content in thoracic aorta. (**c**) Vessel area stained by von Kossa in aortic arch. (**d**) Vascular calcification score calculation was implemented, as described in the Materials and Methods section. (**e**) Correlations were performed between the vascular calcification score and pulse pressure in rats from SNX 0.1 and SNX 0.4 groups. (**f**) mRNA expression was analyzed for genes involved in vascular calcification promotion. (**g**) mRNA expression for α-SMA, and genes associated with vascular calcification inhibition. Bar graphs represent mean ± SEM and *p*-value from Tukey’s multiple comparison test are indicated; ***: *p* < 0.001, **: *p* < 0.01, *: *p* < 0.05. Abbreviations: AA: aortic arch, Alpl: alkaline phosphatase, Bmp2: one morphogenetic protein 2, Enpp1: Ectonucleotide pyrophosphatase/phosphodiesterase 1, Mgp: Matrix Gla protein, Mmp2: matrix metalloproteinases 2, Msx2: Msh homeobox 2, Opn: osteopontin, Pit-1: Solute Carrier Family 20 Member 1, Pit-2: Solute Carrier Family 20 Member 2, Runx2: RUNX family transcription factor 2, SNX: subtotal nephrectomy, TA: thoracic aorta, VC: vascular calcification, α-SMA: Alpha smooth muscle actin.

**Figure 2 cells-12-00643-f002:**
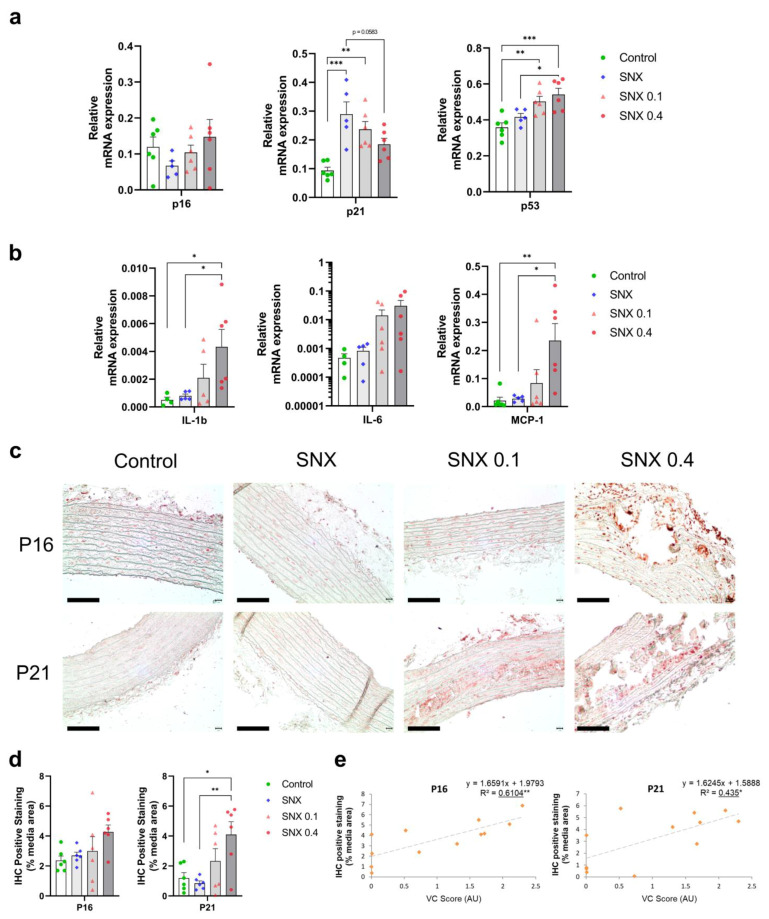
Changes in mRNA and protein expression in markers of cellular senescence and SASP in thoracic aorta of uremic rats. (**a**) mRNA expression of p16^Ink4a^, p21^Cip1^ and p53, markers of cell senescence, in thoracic aorta. (**b**) mRNA expression of pro-inflammatory cytokines contributing to the Senescence Associated Secretory Phenotype (SASP). (**c**) Representative images of immunohistochemistry staining of p16^Ink4a^ and p21^Cip1^ in thoracic aorta. (**d**) Quantification of p16^Ink4a^ and p21^Cip1^ IHC staining. (**e**) Correlation of p16^Ink4a^ and p21^Cip1^ IHC staining with VC score. Bar graphs represent mean ± SEM and *p*-value from Tukey’s multiple comparison test are indicated; ***: *p* < 0.001, **: *p* < 0.01, *: *p* < 0.05. Scale bars correspond to 100 µm. Abbreviations: IHC: immunohistochemistry, IL-1ß: Interleukin 1 beta, IL-6: Interleukin 6, MCP-1: Monocyte Chemoattractant Protein-1, TA: thoracic aorta, VC: vascular calcification.

**Figure 3 cells-12-00643-f003:**
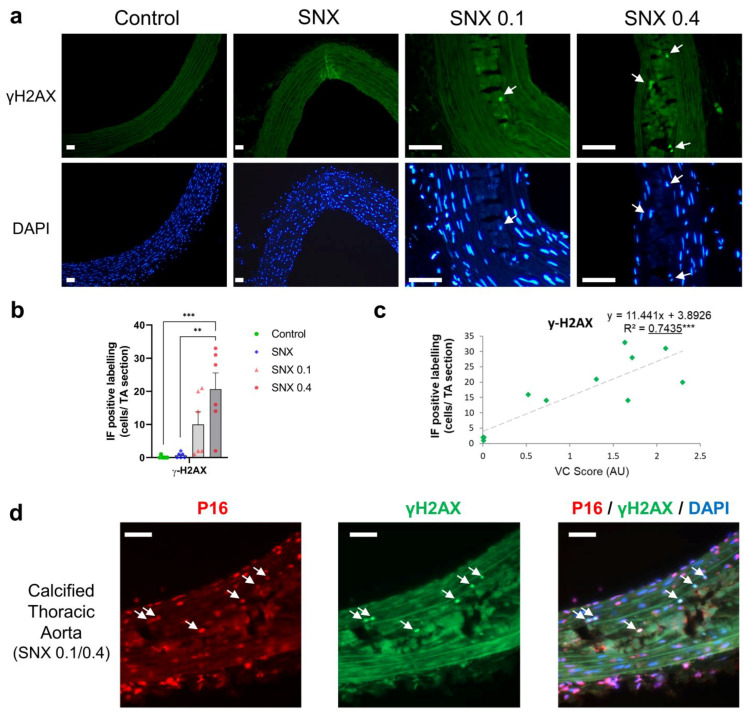
Immunofluorescence of γ-H2AX and co-labeling of γ-H2AX and p16^Ink4a^ in calcified thoracic aortas of uremic rats. (**a**) Representative images of immunofluorescence labeling of γ-H2AX in thoracic aorta of uremic rats and controls. White arrows indicate nuclei with γ-H2AX labeling. The calcified area is highlighted in white. (**b**) Quantification of γ-H2AX-positive cells using immunofluorescence in thoracic aorta sections. (**c**) Correlation between γ-H2AX-positive cells and VC score in SNX 0.1 and SNX 0.4 rats. (**d**) Representative image of p16^Ink4a^ and γ-H2AX co-labeling (white arrows) in calcified thoracic aortas. Bar graph represents mean ± SEM and *p*-value from Tukey’s multiple comparison test are indicated; ***: *p* < 0.001, **: *p* < 0.01. Scale bars = 50 µm. Abbreviations: SNX: subtotal nephrectomy, TA: thoracic aorta, VC: vascular calcification, γ-H2AX: serine 139 phosphorylated H2A histone family member X.

**Figure 4 cells-12-00643-f004:**
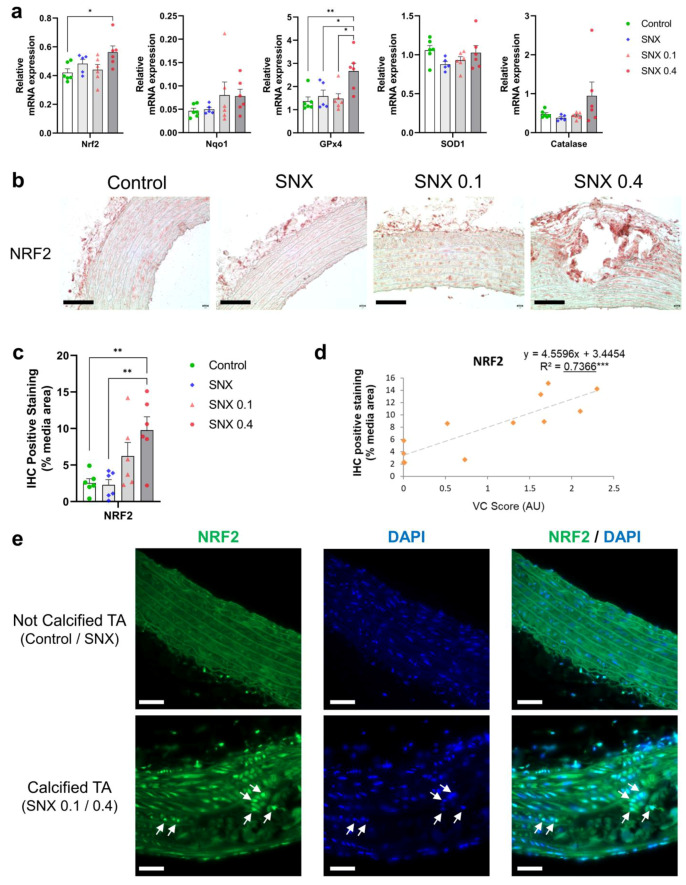
Changes in mRNA and protein expression of NRF2 and associated downstream targets in thoracic aorta of uremic rats. (**a**) mRNA expression of Nrf2 and downstream targets Nqo1, Sod1, GPx4 and Catalase in TA. (**b**) Representative images of NRF2 IHC staining in TA. Black scale bar represents 100 µm. (**c**) Quantification of NRF2 IHC staining in the media of TA. (**d**) Correlation between NRF2 IHC staining and VC score in SNX 0.1 and SNX 0.4 groups. (**e**) Representative images of NRF2 IF labeling in nuclei (white arrows) close to calcium deposits in calcified TA. Labeling was absent in thoracic aorta without calcification (control and SNX groups). White scale bar represents 50 µm. Bar graphs represent mean ± SEM and *p*-value from Tukey’s multiple comparison test are indicated; ***: *p* < 0.001, **: *p* < 0.01, *: *p* < 0.05. Abbreviations: GPx4: Glutathione peroxidase 4, IF: immunofluorescence, IHC: immunohistochemistry, Nqo1: NAD(P)H quinone dehydrogenase 1, NRF2: nuclear factor erythroid 2-related factor 2, SNX: subtotal nephrectomy, TA: thoracic aorta, Sod1: superoxide dismutase type 1.

**Figure 5 cells-12-00643-f005:**
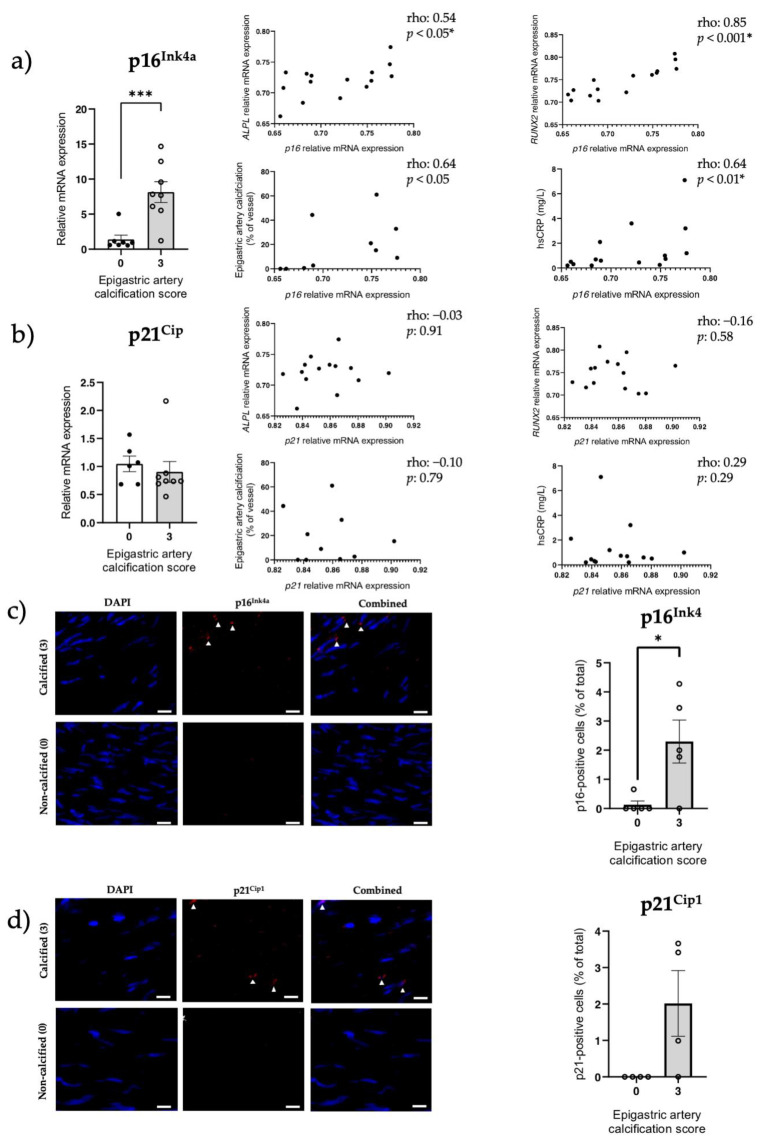
Increased senescent cell burden is associated with presence of calcification in epigastric arteries from kidney failure patients. p16^Ink4a^ and p21^Cip1^ expression, two markers of senescence, were assessed in severely calcified (scored 3) and non-calcified (scored 0) epigastric arteries isolated from kidney failure patients at LD-Ktx. (**a**) p16^Ink4a^ mRNA expression in epigastric arteries (n = 8 per group). Relative gene expression normalized to non-calcified group. p16^Ink4a^ expression was also correlated with *ALPL*, *RUNX2*, epigastric artery calcification as a % and hsCRP. (**b**) p21^Cip1^ mRNA expression in epigastric arteries (n = 8 per group). Relative gene expression normalized to non-calcified group. p21^Cip1^ expression was also correlated with *ALPL*, *RUNX2*, epigastric artery calcification as a % and hsCRP. (**c**) Representative IF images of p16^Ink4a^ labeling in calcified and non-calcified arteries. p16-positive cells are denoted with a white triangle. Quantification of p16^Ink4a^ protein expression in non-calcified (0) and severely calcified groups (3) is also shown. N = 4–5 per group depending on individual vessel integrity after sectioning. Data presented as number of p16^Ink4a^ -positive cells as a % of total number of cells in the media layer of vessels. White scale bar represents 20 µm. (**d**) Representative IF images of p21^Cip1^ labeling in calcified and non-calcified arteries taken from kidney failure patients. p21^Cip1^-positive cells are denoted with a white triangle. Quantification of p21^Cip1^ protein expression in non-calcified (0) and severely calcified groups (3) also shown. N = 4–5 per group depending on individual vessel integrity after sectioning. Data presented as number of p21^Cip1^-positive cells as a % of total number of cells in the tunica media layer of vessels. White scale bar represents 10 µm. Statistics: Spearman rank correlation coefficient is given for non-parametric data. Normality was assessed with Shapiro–Wilk test. Group comparisons were performed using independent t-test for normally distributed data or Mann–Whitney test for non-parametric data. *p*-values < 0.05 were deemed statistically significant. * indicates *p* < 0.05, *** indicates *p* < 0.001. Abbreviations: ALPL: alkaline phosphatase, hsCRP: high-sensitivity C-reactive protein, IF: immunofluorescence, LD-Ktx: living-donor kidney transplantation, RUNX2: RUNX family transcription factor 2.

**Figure 6 cells-12-00643-f006:**
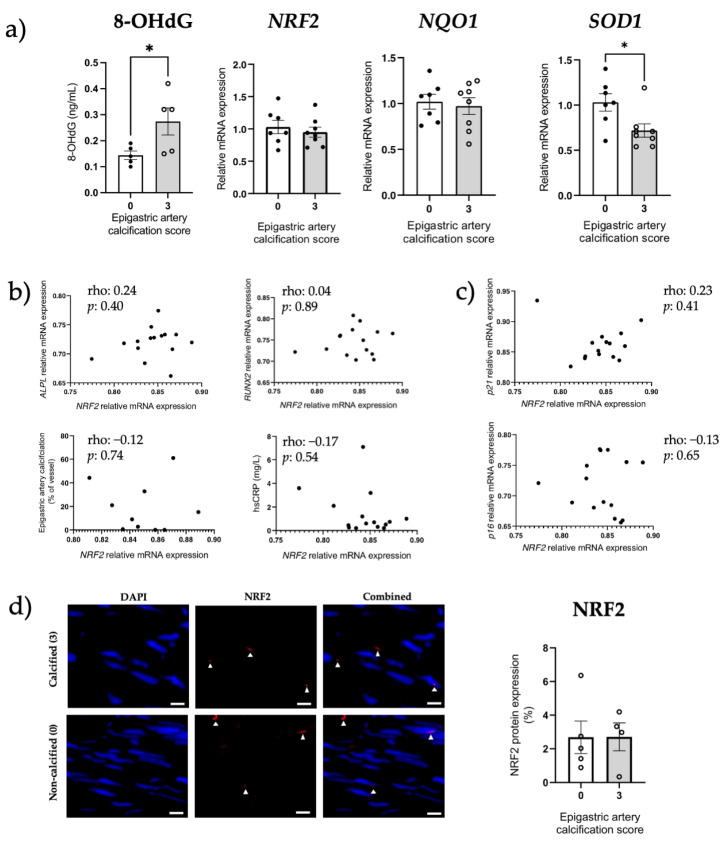
Preserved NRF2 expression between non-calcified and calcified epigastric arteries from renal failure patients. NRF2 expression was assessed in severely calcified (score 3) and non-calcified (score 0) epigastric arteries taken from kidney failure patients at LD-Ktx. (**a**) 8-OHdG measurement, and *NRF2*, *NQO1* and *SOD1* mRNA expression in epigastric arteries (n = 8 per group). Relative mRNA expression normalized to non-calcified group. (**b**) *NRF2* mRNA expression correlated with *ALPL* mRNA expression, *RUNX2* mRNA expression, epigastric artery % calcification and hsCRP. (**c**) *NRF2* mRNA expression correlated with p16^Ink4a^ and p21^Cip1^ mRNA expression. (**d**) Representative NRF2 IF images in calcified and non-calcified arteries. NRF2-positive cells are denoted with a white triangle. Quantification of NRF2 protein expression in non-calcified (0) and severely calcified groups (3) is also shown. White scale bars represent 20 µm. N = 4–5 per group depending on individual vessel structure, expressed as NRF2-positive area as a % of total area of media layer. Statistics: Spearman rank correlation coefficient is given for non-parametric data. Normality assessed with Shapiro–Wilk test. Differences between two groups assessed using unpaired t-test or Mann–Whitney test. *p*-values < 0.05 were deemed statistically significant; * indicates *p* < 0.05. Abbreviations: 8-OHdG: 8-hydroxydeoxyguanosine, ALPL: alkaline phosphatase, hsCRP: high-sensitivity C-reactive protein, LD-Ktx: living-donor kidney transplantation, NQO1: NAD(P)H quinone dehydrogenase 1, NRF2: nuclear factor erythroid 2-related factor 2, RUNX2: RUNX family transcription factor 2, SOD1: superoxide dismutase type 1.

**Figure 7 cells-12-00643-f007:**
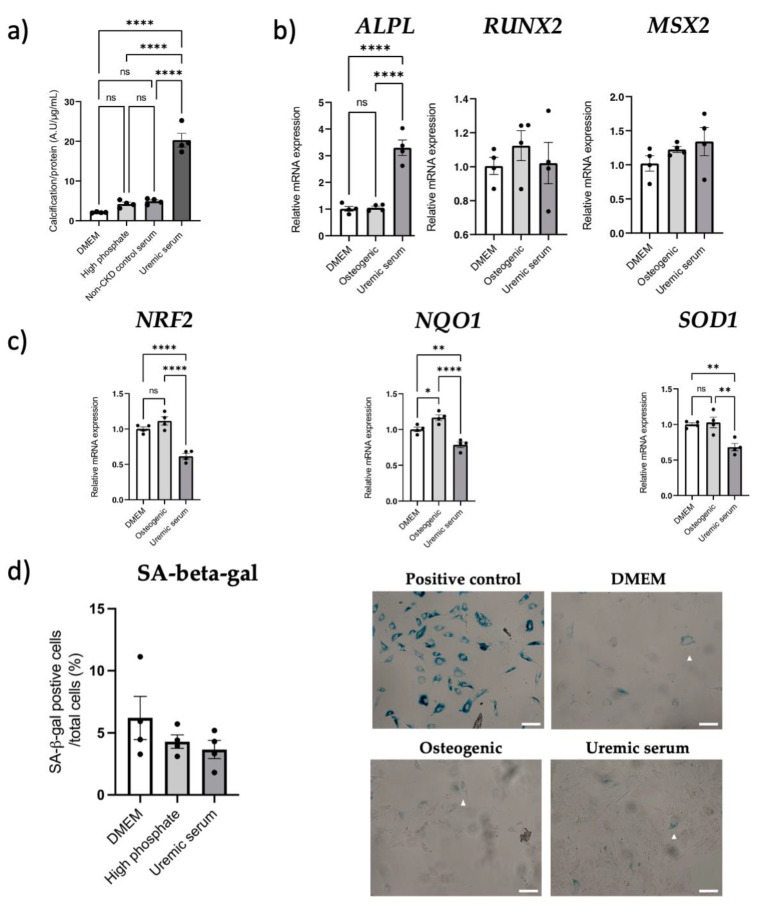
Uremic serum-induced aortic VSMC calcification is associated with dysregulated NRF2, while senescence remains unchanged. Aortic VSMCs were incubated with pooled uremic serum from severely calcified kidney failure patients (n = 8) for 7 days to induce calcification in vitro. Treatment conditions: control DMEM media (DMEM), 2.5 mM phosphate media (high phosphate), 10% pooled non-CKD serum + high phosphate media (non-CKD control serum) or 10% pooled uremic serum + high phosphate media (uremic serum). (**a**) Calcification assay using BoneTag Optical Dye, normalized for protein content (n = 4 per condition). (**b**) mRNA expression analysis of osteogenic markers *RUNX2*, *ALPL*, and *MSX2* (n = 4 per condition). Relative mRNA expression normalized to DMEM control group. (**c**) Gene expression analysis of *NRF2* and downstream genes *SOD1* and *NQO1* (n = 4 per condition), normalized to DMEM control group. (**d**) Quantification and representative images of SA-beta-gal-positive cells (denoted with a white triangle) when incubated with DMEM, osteogenic media or pooled uremic serum + osteogenic media. White scale bar represents 20 µm. Statistics: Groups were compared using one-way ANOVA followed by group-wise comparisons using Tukey’s post hoc test. *p*-values < 0.05 were deemed statistically significant. * indicates *p* < 0.05, ** indicates *p* < 0.01, **** indicates *p* < 0.0001. Abbreviations: ALPL: alkaline phosphatase, Msx2: Msh homeobox 2, NQO1: NQO1: NAD(P)H quinone dehydrogenase 1, NRF2: nuclear factor erythroid 2-related factor 2, RUNX2: RUNX family transcription factor 2, SA-beta-gal: senescence-associated beta-galactosidase, Sod1: superoxide dismutase type 1, VSMC: vascular smooth muscle cell.

**Table 1 cells-12-00643-t001:** Baseline characteristics of the human study cohort, stratified by epigastric artery calcification score (non-calcified vs. severely calcified). Data are presented as mean (standard deviation) for continuous measures, N [percentage] for categorical measures. Differences between strata were assessed by non-parametric Mann–Whitney U test for continuous parameters or Chi-squared test for categorical variables. Significant *p*-values (*p* < 0.05) are depicted in bold. Abbreviations: BMI: body mass index; CAC: coronary artery calcium; eGFR: estimated glomerular filtration rate; hsCRP, high-sensitivity C-reactive protein.

	Non-Calcified Epigastric Artery Group (n = 8)	Severely Calcified Epigastric Artery Group (n = 8)	*p*-Value
Age (years)	47 (15)	54 (13)	0.695
Sex (males/females)	8/0	8/0	-
BMI (kg/m^2^)	23.9 (4.8)	24.1 (4.7)	0.785
Systolic blood pressure (mmHg)	157 (31)	156 (44)	0.242
Diastolic blood pressure (mmHg)	94 (14)	80.0 (27)	0.113
eGFR (mL/min/1.73 m^2^)	8 (2)	10 (4)	0.136
hsCRP (mg/L)	2.1 (2.7)	5.0 (7.9)	0.878
Calcium (mmol/L)	2.3 (0.1)	2.3 (0.4)	0.422
Phosphate (mmol/L)	1.5 (0.4)	1.4 (0.6)	0.798
Cardiovascular disease (yes/no)	0/8	0/8	-
Statin user (yes/no)	6/2	2/6	0.132
Diabetes (yes/no)	1/7	2/6	1.000
CAC score (AU)	0 (0)	463 (1165)	**0.018**

**Table 2 cells-12-00643-t002:** Blood and urine parameters related to renal function, phosphocalcic metabolism, hemodynamic and cardiac parameters in uremic and control rats. Sprague Dawley rats underwent 5/6th nephrectomy (SNX) in a one-step procedure and six rats underwent a sham operation (control group). Rats were then placed on a high-phosphate and high-vitamin D diet for four weeks (0.1 µg/day: SNX 0.1 group; 0.4 µg/day: SNX 0.4 group). The SNX group was only subjected to the SNX. Arterial pressure was measured under anesthesia after catheterization of the right carotid of the animals with a pressure transducer connected to a computer. Carotid catheterization failed for one rat in the SNX group; arterial pressure was not measured and blood was not obtained for this animal. Creatinine and urea concentrations were assessed in plasma and urine. Calcium and phosphate concentrations were assessed in plasma. Daily urine volume was determined after the rats were maintained in metabolic cages for 3 days and creatinine clearance was calculated. Protein in urine was also measured. The heart was harvested and the left ventricle was isolated and weighed. Heart and kidney fibrosis was determined histologically by Sirius red staining. Data presented as mean ± SEM. Data in bold followed by * are significantly different from the control group (the *p*-value from Tukey’s multiple comparison test are indicated; ***: *p* < 0.001, **: *p* < 0.01, *: *p* < 0.05, *t: 0.05 < *p* < 0.1). Data in bold followed by # are significantly different from the SNX 0.4 group (#: *p* < 0.05, #t: 0.05 < *p* < 0.1). The *p*-value from ANOVA test is indicated in the far right column (in bold if *p* < 0.05). Abbreviations: Ca: Calcium, P: Phosphate, SNX: Subtotal Nephrectomy.

	Control (n = 6)	SNX (n= 5)	SNX 0.1 (n = 6)	SNX 0.4 (n = 6)	*p*-Value
Final Weight (g)	491 ± 24	530 ± 39	538 ± 21	474 ± 25	0.3026
Arterial Pressure					
Systolic Arterial Pressure (mmHg)	155 ± 8	190 ± 18	161 ± 16	184 ± 13	0.2502
Diastolic Arterial Pressure (mmHg)	118 ± 8	133 ± 11	105 ± 12	97 ± 9	0.0983
Pulse Pressure (mmHg)	38 ± 1	57 ± 7 #t	**56 ± 9 #**	**87 ± 10 *****	**0.0017**
Hematocrit (%)	45 ± 1	38 ± 2	39 ± 3	36 ± 3	0.0533
Plasma					
Creatinine in Plasma (µM)	26 ± 1	91 ± 20	94 ± 18	104 ± 34	0.0675
Urea in Plasma (µM)	5.6 ± 0.2	**14.4 ± 2.9 ***	**14.4 ± 3.0 ***	11.7 ± 1.8	**0.0253**
Calcium in Plasma (mM)	2.51 ± 0.02	2.72 ± 0.04	2.77 ± 0.06	**2.85 ± 0.12 ***	**0.0284**
Phosphate in Plasma (mM)	1.79 ± 0.03	2.11 ± 0.11	2.54 ± 0.32	2.88 ± 0.43	0.0559
Product [Ca × P] in Plasma (mM)	4.50 ± 0.12	5.73 ± 0.32	7.11 ± 1.02	**8.07 ± 0.97 ***	**0.0264**
Urine					
Creatinine in Urine (mM)	5.20 ± 0.62	**2.33 ± 0.73 *t**	**2.26 ± 0.49 ***	3.29 ± 0.94	**0.029**
Urea in Urine (µM)	329 ± 30	**136 ± 48 ****	**132 ± 23 ****	**179 ± 35 ***	**0.0015**
Daily Urine Volume (mL/24 h)	14 ± 1	**26 ± 4 ***	**29 ± 4 ****	24 ± 2	**0.009**
Proteins in Urine (mg/24 h)	6.5 ± 1.1	60.0 ± 21.2 *t	60.6 ± 8.0 *t	50.8 ± 20.4	**0.0494**
Creatinine Clearance (mL/min)	1.87 ± 0.19	**0.47 ± 0.14 ****	**0.58 ± 0.20 ****	**0.83 ± 0.36 ***	**0.0026**
Kidney Fibrosis (% area stained)	4.7 ± 0.7	8.2 ± 1.2	10.3 ± 2.7	10.6 ± 2.1	0.1724
Left Ventricle weight (g)	1.00 ± 0.05	**1.57 ± 0.07 *** #**	**1.54 ± 0.05 *** #**	**1.33 ± 0.03 ****	**<0.0001**
Left Ventricle weight (% of body weight)	0.21 ± 0.01	**0.30 ± 0.02 ****	**0.29 ± 0.02 ****	**0.28 ± 0.02 ***	**0.0019**
Heart Fibrosis (% area stained)	2.7 ± 0.3	3.3 ± 0.8	**2.8 ± 0.5 #**	**8.6 ± 2.7 ***	**0.0242**

**Table 3 cells-12-00643-t003:** Correlation between vascular calcification score (VC Score) and mRNA expression of multiple markers in thoracic aorta of SNX 0.1 and SNX 0.4 rats. We correlated the mRNA expression of multiple markers in thoracic aorta of SNX 0.1 and SNX 0.4 rats, presenting different levels of calcification, with VC score. Values of Pearson’s correlation coefficients (r) and *p*-values are given (in bold if *p* < 0.05). Visual representations of correlations are presented in Appendix A. Abbreviations: SASP: Senescence Associated Secretory Phenotype, VC: Vascular Calcification.

	r	*p*-Value
VC regulation and phenotypic transition		
*Opn*	0.873	**0.0002**
*a-SMA*	−0.837	**0.0007**
*Alpl*	−0.820	**0.0011**
*Mmp2*	−0.807	**0.0015**
*Runx2*	0.763	**0.0039**
*Mgp*	0.737	**0.0062**
*Pit-1*	0.731	**0.0069**
*Pit-2*	−0.658	**0.0198**
*Enpp1*	0.637	**0.0351**
*Msx2*	0.600	**0.0393**
*Bmp2*	0.596	**0.0412**
*Tgf-β1*	0.402	0.1962
Markers of senescent cells		
*p16 ^Ink4a^*	0.669	**0.0174**
*p21^Cip1^*	−0.575	0.0506
*p53*	−0.041	0.8984
Cytokines contributing to the SASP		
*Mcp-1*	0.640	**0.0250**
*IL-1b*	0.546	0.0751
*TNFa*	0.390	0.2096
*IL-6*	0.254	0.4259
NRF2 pathway		
*Nrf2*	0.663	**0.0186**
*Nqo-1*	0.813	**0.0013**
*SOD1*	−0.462	0.1307
*Catalase*	−0.293	0.3559
*GPx4*	−0.111	0.7312

## Data Availability

The datasets generated and analysed during the current study are available from the corresponding author on reasonable request.

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
