# Peer review of "Implications of Senescent Cell Burden and NRF2 Pathway in Uremic Calcification: A Translational Study"

_cells, 2023, doi:10.3390/cells12040643_

Round 1
Reviewer 1 Report
The manuscript “Implications of Senescent Cell Burden and NRF2 Pathway in Uremic Calcification: A Translational Study” by Jonas Laget et al investigate mechanisms whereby senescence and NRF2 dependent genes are implicated in the vascular calcification process.
The study supplies a large amount of new information in the field of vascular calcification. The data is obtained from very different methodological approach that to some extent are complementary.
I have few comments, questions and suggestions:
-The authors stated several times in the manuscript that they use a new model of vascular calcification in rats: 5/6Nx fed a high P diet receiving supplement of vitamin D. I have seen this rat model published in several studies. (Am J Nephrol 2010; 31:471–481)
- Information on the Calcium content or the rat diet should be included
- Table 1 should include serum, Calcium and Phosphate levels (if the data is available)
- Legend of Table 2 should explain what is the meaning of SNX, and 0.1 or 0.4
- Alpha-SMA in legend and a-Sma in the figure 1.
- The authors should go over the text to correct some errors errors
-It is clear that a significant effect resulting from senescent cells is the senescence-associated secretory phenotype (SASP), with significant increase expression and secretion of diverse pro-inflammatory cytokines. This is an important issue that the author may want emphasize. Vascular calcification is not only the result of inflammation but also the cause of inflammation (and of course a large number of concomitant factors).
-The differences between P16 and P21 are rather interesting because, as mentioned in the manuscript, the roles in the development of cell senescence are different. p21 may play a role in inactivation of the DNA replication factor proliferating cell nuclear antigen during early senescence; p16 may be upregulated as part of a differentiation program that is turned on in stablished senescent cells. Therefore if p21 decreases after senescence is achieved, upregulation of p16 may be essential for maintenance of the senescent-cell-cycle arrest. (Mol Cell Biol. 1999 Mar; 19(3): 2109–2117)
In figure 5a the correlation Runx with p16 is almost perfect. Could we assume that to have osteogenic phenotype cells must be senescent ?
In figure 6c the correlation Runx2 vs p21 is not significant but I see that the correlation is acceptable except for un sample that appear to be from another group . I would be interesting to hypothesize that NRF2 expression occurs during early process of senescence and it disappear when senescence is established. Any way , it may be worth to correlate P21 with p16
Reviewer 2 Report
Reviewer(s)' Comments to Author:
Reviewer: 1
The authors of this study analysed the implication of senescence and NRF2 pathways in uraemia-induced vascular calcification using three complementary approaches. The authors used (a) the model of 5/6 nephrectomised rats supplemented with high phosphate and vitamin D, (b) calcified epigastric arteries (EAs) of chronic renal failure patients and (c) vascular smooth muscle cells (VSMCs) incubated with uremic serum from kidney failure patients with vascular calcifications. Vascular calcification processes, presence of senescent cells and the relevance of the NRF2 pathway were analysed. The authors were able to demonstrate an increased expression of p16 and p21, as well as y-H2AX positive cells highlighting an increased senescent cell burden in the in vivo model, but calcification of vascular smooth muscle cells induced by uremic serum was not accompanied by senescence. NRF2 levels and corresponding downstream genes were differently regulated in the three models highlighting context-dependent adaptations. The authors concluded that the modulation of the NRF2 pathway is a potential therapeutic option to combat vascular calcification in chronic renal failure patients. The approach of the study is of great interest, and answering the following comments will improve the context and impact of the study and its results:
Minor points:
- The definition and determination of CKD classification used in the manuscript should be explained in more detail.
- Vitamin K, as well as inositol phosphates and other mediators, were recently identified as inhibitors of vascular calcification processes. The authors should mention these mediators in the introduction of the revised version of the manuscript.
- The rationale for the number of experiments should be described in the manuscript.
- Can the authors explain more in detail why only male subject were included in the study?
- Could the authors provide the secondary antibody-only control performed for IHC experiments (Figures 2 and 4).
- The discussion of the underlying mechanisms could be expanded.
Reviewer 3 Report
The authors aimed to investigate the role of senescent cell burden and dysregulation of the nuclear factor erythroid 2–related factor 2 (NRF2) pathway in promoting vascular calcification (VC) in chronic kidney disease (CKD) in this study.
The model created to show this effect is quite impressive.
The authors investigated whether senescence and NRF2 pathways may serve as drivers of uremia induced VC using three complementary approaches:
1. a novel model of induced VC in 5/6-nephrectomized rats supplemented with high phosphate and vitamin D;
2. epigastric arteries (EAs) from CKD patients with established medial calcification;
3. and vascular smooth muscle cells (VSMCs) incubated with uremic serum.
The authors observed that expression of p16 and p21, as well as y-H2AX positive cells, confirmed increased senescent cell burden at the site of calcium deposits in aortic sections in rats, and was similarly showed in calcified EAs from CKD patients. However, they also noticed that uremic serum-induced VSMC calcification was not accompanied by senescence. Interestingly the authors observed that expression of NRF2 and downstream genes, Nqo1 and Sod1, associated with calcification in uremic rats, while there was no difference between calcified and non-calcified EAs. Conversely, in vitro uremic serum-driven VC was associated with depleted NRF2 expression.
At the end of the study, they concluded that their data supports existing data that shows senescent cells may promote VC, while NRF2 regulation differs between different models of uremia-induced VC, strengthening the importance of both pathways as potential therapeutic options to combat VC in CKD patients.
It was a good study in terms of revealing the pathologies that occur with the dysregulation of the NRF2 pathway or in which the NRF2 pathway is effective.
The introductory part of the article explained very well the background information and why it was necessary to do this study.
In the material and method section, it is stated how the modeling is done and how the expected statistics are reached.
The tables and figures given in the results section are extremely high quality and explanatory.
The discussion is well designed. The comments are descriptive and sufficient. References are well written. The data obtained as a result of this study are at a level that can shed light not only on the pathogenesis of vascular calcification, but also on new studies.
In this context, I have no criticism or additional contribution. Congratulations to the researchers, I liked it very much.
Round 2
Reviewer 1 Report
The authors have addressed all comments and question appropriately
Reviewer 2 Report
All my comments and suggestions have been very well addressed in the revised form of the manuscript. I recommend the acceptance of the manuscript.